



# Using a data-driven statistical model to better evaluate surface turbulent heat fluxes in weather and climate numerical models: a demonstration study

Maurin ZOUZOUA [1], Sophie BASTIN [1], Marjolaine CHIRIACO [1], Fabienne LOHOU [2], Marie LOTHON [2], Mathilde JOME [2], Cécile MALLET [1], Laurent BARTHES [1], and Guylaine CANUT [3]

[1]Laboratoire Atmosphère, Observations Spatiales (LATMOS)/Institut Pierre Simon Laplace (IPSL)
[2]Centre de Recherches Atmosphériques (CRA)/Laboratoire d'Aérologie de Toulouse (LAERO)
[3]Centre National de Recherches Météorologiques (CNRM)/Météo-France

**Correspondence:** (maurin.zouzoua@latmos.ipsl.fr)

**Abstract.** This study proposes the use of a data-driven statistical model to freeze the errors due to differences in environmental forcing when evaluating the surface turbulent heat fluxes from weather and climate numerical models with the observations. It takes advantage of continuous acquisition over approximately ten years of near-surface sensible and latent heat fluxes ($H$ and $LE$ respectively) together with ancillary parameters over the supersite "Météopole" of the French national research infras-

5 tructure ACTRIS-FR, located in Toulouse. The statistical model consists of several multi-layer perceptrons (MLPs) with the same architecture. Thirteen variables characterizing the environmental forcing in the surface layer at an hourly time scale are used as input parameters to estimate $H$ and $LE$ simultaneously. The MLPs are trained using 5-year observational data under a 5-fold cross-validation. The remaining data is used to test the estimates on unknown conditions. A case study is performed with data from a regional climate simulation. The performance of the statistical model ranges within the state-of-the-art surface

parametrization schemes on hourly and seasonal time scales. It has also a good generalization ability, but hardly estimates negative $H$ and large $LE$. The statistical model is used to evaluate the simulated fluxes under the simulated environment to better examine the flaws of their numerical formulation throughout the simulation. Comparison of simulated fluxes with observed and MLP-based fluxes show different results. According to MLP-based fluxes in the simulated environment, the land surface scheme of this climate model tends to underestimate large sensible heat flux. Thus, it incorrectly partitions between surface heating and evaporation during the late summer. Our innovative method provides insight into differently evaluating the simu-

lated near-surface turbulent heat fluxes when a long period of comprehensive observations is available. It can usefully support ongoing efforts for improvements of surface parametrization schemes.





## 1 Introduction

The surface sensible heat ($H$) and latent heat ($LE$) fluxes describe the surface-atmosphere exchanges of heat and moisture
(Stull, 1988). They are major terms of the Surface-Energy-Budget (SEB) and key drivers of the atmospheric boundary layer
(ABL) process, such as turbulent mixing and convective cloud formation. Numerical models are important tools for weather
forecasting and climate projection. Due to the coarse spatio-temporal resolution of operational weather and climate numerical
models, the surface turbulent heat fluxes are computed with the help of surface parametrization schemes, which have different
levels of sophistication. The correct representation of turbulent heat fluxes by these schemes is then necessary for properly
simulating the surface-atmosphere interactions. However, this representation is the second most important source of biases in
simulations with the numerical models (Zadra et al., 2018). It is therefore of paramount importance to develop improvements,
and evaluation is crucial to provide guidance.

The strategies for evaluating surface parametrization schemes can be roughly classified into two main approaches (Henderson-
Sellers et al., 1996). The first involves running full numerical simulations, with meteorological forcing and surface turbulent
heat fluxes interacting mutually. The simulated surface turbulent heat fluxes are then confronted with observations. This ap-
proach blends many sources of errors such as inconsistent landscape representation (vegetation and soil characteristics) and
inaccurate weather conditions (cloudiness, temperature, moisture, wind, etc.). It is likely a useful method for assessing how
well a numerical model is working. However, the formulations that resolve the surface-atmosphere interactions cannot be un-
ambiguously evaluated since the simulated turbulent fluxes are related to simulated weather conditions that are not necessarily
observed. Various strategies using full numerical simulation have been proposed to reduce the sources of errors, for example
by investigating the relationships between surface heat fluxes and driving atmospheric variables (e.g., Zhou and Wang, 2016;
Bastin et al., 2018). Nevertheless, there are still large uncertainties. The second approach suppresses the errors associated with
the simulated weather conditions by externalizing the surface scheme to the numerical model. The turbulent fluxes are then
computed thanks to input from observations or reanalysis. However, the crucial influence of turbulence fluxes on weather con-
ditions is not taken into account. Moreover, the representativity of surface characteristics remains problematic because several
required properties (roughness length, soil and vegetation parameters, etc.) are not usually observed. The use of their default
or empirical values is an additional source of uncertainties (Liu et al., 2013). The intrinsic limitations of these two approaches
demonstrate the need for another to reliably evaluate the numerical formulation of turbulent heat fluxes.

In recent years, machine learning techniques have known a tremendous expansion in weather and climate sciences (de Burgh-
Day and Leeuwenburg, 2023), driven by unrivalled results and infinite possibilities. Due to their ability to act as universal



approximators (Cybenko, 1989; Hornik et al., 1989), Artificial Neural Networks (ANNs) have emerged as a powerful tool in machine learning (Goodfellow et al., 2016) for data-driven statistical modelling. They can effectively model a broad range of complex relationships for quantitative modellings, such as multivariate classification and regression (Zhang, 2008; Kruse et al., 2013). ANNs are generally used to overcome the limitations of classical approaches. Several studies have explored the use of ANN-based estimators for replacing numerical atmospheric models or some of their components (e.g., Bonavita and Laloyaux, 2020; Gentine et al., 2018; Knutti et al., 2003; Sarghini et al., 2003; Vollant et al., 2017). In the study of Abramowitz (2005), a trained ANN with observational data is used as a benchmark to objectively assess how well a land surface scheme should perform in estimating turbulent heat and net $CO_2$ fluxes. Recently, Leufen and Schädler (2019) estimated the scaling quantities needed in some surface parameterization schemes to calculate the momentum and sensible heat fluxes using an ANN-based model driven by meteorological factors. The ANN has learned from multi-year comprehensive data collected over several types of landscapes (grassland, forest, etc). They obtained satisfying results when this ANN was implemented to replace the similarity functions in a one-dimensional stand-alone land surface model. In the field of hydrology, ANN-based models are increasingly being employed to estimate reliable evapotranspiration for near real-time monitoring of crop water demand (Kumar et al., 2011; Kelley et al., 2020; Kelley and Pardyjak, 2019). The growing availability of comprehensive data from atmospheric observatories offers an opportunity to explore ANN-based methods to better evaluate the numerical formulation of $H$ and $LE$, particularly within the framework of full simulations, which is the ultimate goal of numerical models.

The Model and Observation for Surface-Atmosphere Interactions (MOSAI) project (Lohou et al., 2022) aims to contribute to a better understanding of surface-atmosphere interactions to improve their representation in weather and climate numerical models. One of the principal objectives is to propose a novel method to diagnose the errors of numerical models in their formulations of $H$ and $LE$. The idea is to exploit the capabilities of machine learning techniques on multi-years of continuous observational data rather than performing a direct comparison of simulated against observed fluxes. This paper is a contribution to this objective. It presents a pilot study that uses data collected during several years at a French flux station, operational since June 2012, for evaluating turbulent heat fluxes in a climate numerical model over the period from 01 January 2012 to 31 December 2016. Section 2 presents the proposed evaluation approach, which involves using observational data to build a data-driven statistical model that estimates $H$ and $LE$. The data and methods of our experimental setup are described in section 3. Section 4 discusses the performance of the data-driven model in observed conditions. In the section 5, the data-driven model is applied to simulated conditions to better identify the flaws in the numerical formulation of turbulent heat





fluxes. Finally, section 6 delivers a conclusion.

## 2 Justification and objectives

The fluxes $H$ and $LE$ are primarily governed by the net radiative flux ($R_{net}$) at the surface which is the algebraic sum of
incoming ($\downarrow$) and outgoing ($\uparrow$) longwave (LW) and shortwave (SW) radiations. Their magnitude is as well determined by
thermodynamical and dynamical conditions within the surface layer; a thin atmospheric layer immediately above the ground
in which the turbulent fluxes are approximately constant. The flux $H$ is responsible for removing/depositing heat from/to the
ground, and $LE$ is the energy exchanged through phase changes of water from liquid (or ice) to vapour. $H$ and $LE$ are therefore
closely linked to the vertical gradients of temperature and humidity in the surface layer. The relative predominance between $H$
and $LE$ depends on surface characteristics (vegetation and soil moisture), $LE$ is predominant over wet surfaces and vice-versa.
Solar heating and the intra-annual evolution of land cover induce diurnal and seasonal cycles of turbulent heat fluxes. Thus, the
turbulent fluxes result from complex non-linear relationships between meteorological factors, surface cover and soil conditions.

Current numerical formulations of turbulent heat fluxes are largely based upon the validity of the Monin–Obukhov similarity
theory (MOST, Monin and Obukhov, 1954) in the surface layer, which relies on horizontally homogeneous terrain, fair and
steady-state meteorological conditions. The fluxes are then expressed in terms of the vertical gradient of the corresponding
quantity (temperature and humidity for $H$ and $LE$ resp.) within this layer and, various parameters describing soil wetness and
roughness. The weather and climate models usually apply MOST between the ground and the lowest atmospheric level (Liu
et al., 2013). Another fundamental relationship, that is usually used in these models, is the conservation of SEB as follows:

$$R_{net} = H + LE + G \tag{1}$$

Where $G$ is the ground heat flux. However, this conservation is rarely verified when $H$ and $LE$ are measured with an Eddy-
Covariance (EC) method, the most recent and reliable technique (Mauder and Foken, 2004; Wolf et al., 2008; Aubinet et al.,
2012). Indeed, the available energy $R_{net} - G$ is very often greater than the total turbulent flux $H + LE$, especially over a
heterogeneous surface (Hu et al., 2021; Mauder et al., 2018; Foken et al., 2011). This imbalance can be quantified by the
residual energy (RES).



$$RES(\%) = 100 \cdot \frac{(R_{net} - G) - (H + LE)}{R_{net} - G} \tag{2}$$

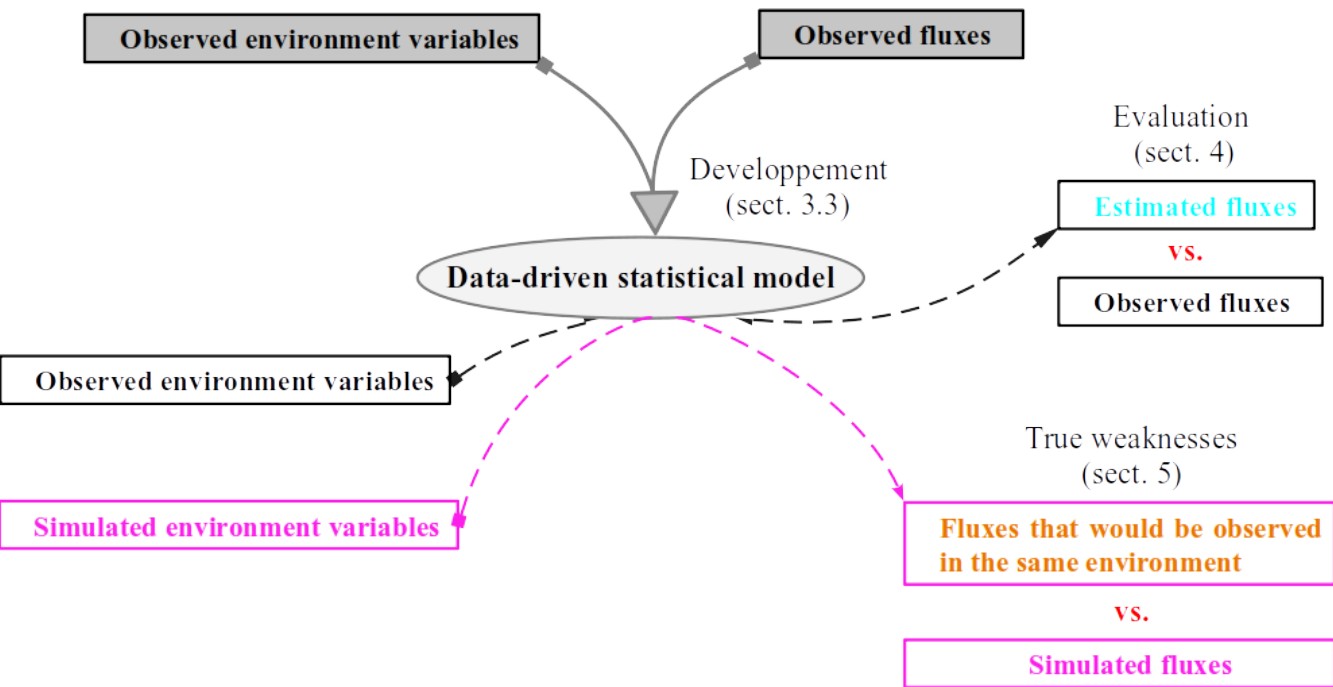

**Figure 1.** Schematic illustration of our proposed evaluation method.

Thus, the discrepancy between simulated and observed $H$ and $LE$ may arise, on the one hand, to incorrect parametrizations
and, on the other hand, due to both observation and model biases. The direct comparison is therefore less useful to point out
the errors inherent to numerical formulations of $H$ and $LE$. The study by Abramowitz (2005), based on observational data,
is a pioneer in using a data-driven statistical model to reliably assess land surface schemes. Inspired by the methodology
of this study, we propose an evaluation approach dedicated to a full numerical simulation that includes a more realistic
representation of the interplay between surface heat fluxes and environmental factors. It consists of two successive phases
illustrated in Figure 1. At first, a long period of $H$ and $LE$ observations together with related environmental factors is required
to build a data-driven statistical model estimating $H$ and $LE$. It can be regarded as a parametrization without any simplifying
assumptions. Then, the application of this statistical model to the simulated environment will generate the fluxes that would

have been observed from a statistical point of view. Thus, by comparing simulated heat fluxes with their statistically-based estimates corresponding to the simulated environment, the uncertainties due to model biases are frozen. This allows us to
better diagnose weaknesses in the $H$ and $LE$ formulations or due to surface parameters and characteristics.

## 3 Experimental setup

### 3.1 Observational data

This pilot study is based on high-temporal-resolution data gathered over several years at the Météopole (43.57°N, 1.374°E,
157 m above sea level, Etienne, 2022), a measurement site hosted by the Centre National de Recherches Météorologiques (CNRM) in Toulouse, France. This site, operated by Meteo-France, is part of the Aerosol, Clouds and Trace Gases Research Infrastructure (ACTRIS). The observation facility consists of several co-located ground-based instruments installed in a large grass field. The AERIS platform (https://www.aeris-data.fr/) provides free access to this dataset.

Since 15 June 2012, a comprehensive set of parameters documenting soil conditions, overlying meteorological forcing and surface energy budget is collected. The most relevant ones for our study are listed in Table 1. These include soil heat flux $G$, the four radiative flux components, dry air temperature ($T$) and relative humidity ($RH$) at the two conventional heights 2 and 10 m above ground level (agl), surface pressure ($SP$) and rainfall ($RR$), and soil volumetric water content ($SWC$). They are originally acquired every 1-min and finally archived as half-hourly averages. These data undergo several quality checks before
being released to the public. The surface and soil temperature are also measured, but there is a lack of data before 12 July 2015. We then opted not to use these measurements to avoid limiting the number of samples, which is crucial when building a data-driven statistical model.

The near-surface sensible ($H$) and latent ($LE$) heat fluxes are estimated with the EC method (Aubinet et al.,
2012) based on high-frequency measurements (20 Hz) of the wind 3D components, $T$ and water vapour specific humidity ($q$) at 3.7 magl, with sonic anemometer and rapid hygrometer respectively. EddyPro 7 software (https://www.licor.com/env/support/EddyPro/software.html) is used to compute the fluxes over half-hourly samples. Quality flags, defined according to Mauder and Foken (2004), rank the measurements into three different categories: 0 for the best quality, 1 for the suitable and 2 for those that should not be used for analysis. Besides, the measurements of the used





rapid hygrometer (licor 7500 open-path) are highly degraded in wet conditions (fog and rain). Thus, the turbulent fluxes are normally not estimated in these conditions, based on the sensor diagnostic and the rainfall occurrence. Jomé et al. (2023) analyzed the contribution of the surrounding land cover types to the turbulent heat fluxes measured at the Météopole-flux station. It was found that the contribution of grass cover ranges between 80 to 90 %, with the remaining contribution coming mostly from urbanized areas.


**Table 1.** Observational data from the Météopole flux station used in this study. A negative height corresponds to a soil depth.

| Variables | Height (m agl) |
|---|---|
| Surface upward/downward long/short-wave components | 10 |
| Turbulent heat fluxes ($H$ and $LE$) and horizontal wind components ($u$, $v$) | 3.7 |
| Relative humidity ($RH$) and dry air temperature ($T$) | 2 and 10 |
| surface pressure ($SP$) and rainfall ($RR$) | - |
| Heat flux into the ground ($G$) | $-0.05$ |
| Soil volumetric water content ($SWC$) | at 16 levels, the first at $-0.1$ |

The data availability for the turbulent heat fluxes and ancillary parameters overlap continuously from $00:00$ UTC 24 November 2012. At the time of this study, the data collected up to $23:30$ UTC 31 December 2022 have been released. The corresponding database contains 117778 half-hourly time steps for which turbulent heat fluxes, meteorological and soil parameters are simultaneously available, i.e nearly 66.5 % of the samples expected since $00:00$ UTC 24 November 2012.

This percentage is highly affected by the lack of turbulent fluxes during wet conditions. Since the quality and amount of the data on which the data-driven statistical model is built determines its performance, several considerations were applied to select the most reliable measurements of $H$ and $LE$. Firstly, only $H$ and $LE$ with quality flag 0 or 1 (Mauder and Foken, 2004) were selected. Due to the strong evolution of the surface heat fluxes along the day, several authors preferred to evaluate the numerical simulations under well-established diurnal cycles (e.g. Román-Cascón et al., 2021). Then, secondly,

we considered sampling at a daily scale when selecting the final observational data. The half-hourly data included in our analyses are collected during the diurnal cycles (starting at $00:00$) that fulfil the three following conditions: (i) it is described by at least half of the expected samples (i.e 24/48 of pre-selected samples), (iii) the daily cumulative rainfall is less than 5 mm and (iii) all the items of Pearson's correlation coefficient matrix between $H$, $LE$ and SW$^\downarrow$ are greater than 0.6. The two last criteria are a compromise to preserve a reasonable number of samples while reducing the amount

of data possibly impacted by wet conditions. This leaves us with 66273 half-hourly samples (about 56% of all available samples), documenting 1983 diurnal cycles. Figure 2a shows the distribution of these samples per year and Figure 2b presents the distribution of the corresponding diurnal cycles per month. None of these diurnal cycles was fully sampled,



the rate of data availability is around 70 % on average. The diurnal cycles are overall homogeneously distributed over the year. The four seasons (winter, spring, summer and autumn) are therefore rather well covered. The annual cycles from 2015

to 2021 are particularly well sampled. There are no selected samples from June to September 2022 due to missing data of LW$^\uparrow$.

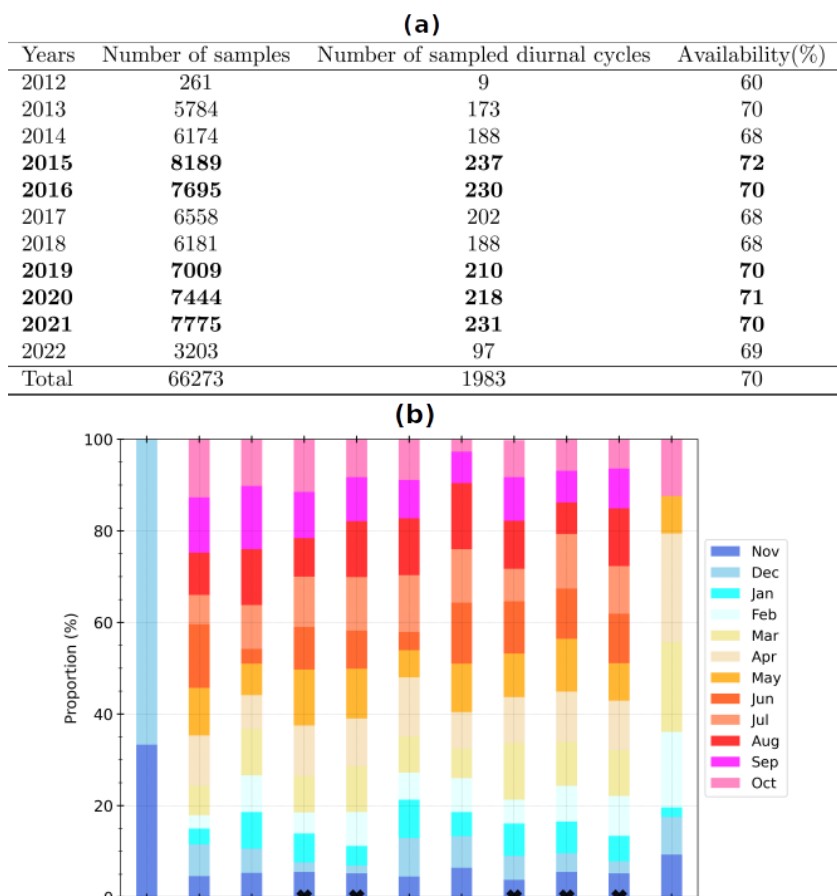

**Figure 2.** (a) Number of remaining half-hourly samples and diurnal cycles per year after the selection over the period from 24 November 2012 at 00 : 00 to 31 December 2022 at 23 : 30 (see text for details), and (b) monthly distribution of these diurnal cycles. The rate of availability is the ratio of the selected half-hourly samples (second column) over the number of samples which fully describe the corresponding diurnal cycles (48 times the third column). The data of the five most covered years (in bold (a), with a black cross at the bottom (b)) composed the learning set, the other years are used as the test set.



## 3.2 Numerical model data

To test our proposed evaluation method, we used data from an existing climate simulation, carried out with the Regional Earth system model of the Institut Pierre Simon Laplace (RegIPSL). Within the settings of this model, the land surface model ORCHIDEE (ORganising Carbon and Hydrology In Dynamic EcosystEms, Krinner et al., 2005) provides the bottom boundary conditions to the atmospheric model WRF (Weather Research and Forecasting, Skamarock et al., 2008) over the continental surface. The simulation has been performed in the framework of the Mediterranean Coordinated Regional Climate Downscaling Experiment (Med-CORDEX) initiative (Ruti et al., 2016) and the European Climate Prediction system (EUCP) H2020 project (Coppola et al., 2020). It covers the Euro-Mediterranean area with a horizontal resolution of $20\,\mathrm{km}$ on a Lambert-conformal projection and spans from $1^{st}$ January 1979 to 31 December 2016. The atmospheric vertical column was discretized by $46$ hybrid sigma-pressure levels (full eta-levels), with 16 levels roughly in the first $2\,\mathrm{km\,agl}$. The soil column, which extends until $2\,\mathrm{m}$ below the ground, was subdivided by 11 nodes. Seven of these nodes were located in the top $15\,\mathrm{cm}$. For more details, readers may refer to the studies of Guion et al. (2022) which used this climate simulation to assess the impact of droughts and heatwaves upon vegetation and wildfires in the Western Mediterranean, and of Shahi et al. (2022) which used the RegIPSL model to analyse the added-value of a convective-permitting climate simulation over the Iberian peninsula.

The landscape in ORCHIDEE was categorized into 13 main classes including bare soil and 12 Plant Functional Types (PFTs: eight for forests, two for grasslands and two for croplands). The total proportion of the grid cell occupied by each class remained constant throughout the simulation. Nonetheless, for PFTs, the proportion effectively occupied by vegetation was allowed to vary and the non-occupied fraction was defined as bare soil (Ducharne et al., 2018; Alléon, 2022). The bare soil fraction is assumed to contain the urbanized areas. For simulating the surface processes, ORCHIDEE requires several environmental parameters including surface precipitation, downwelling $SW$ and $LW$ as well as air temperature, humidity and wind just above the ground. These were taken at the lowest vertical level of WRF, which was located within $20\,\mathrm{m\,agl}$. The near-surface turbulent heat fluxes are computed using bulk aerodynamic formulations (Ducoudré et al., 1993; Krinner et al., 2005; Alléon, 2022) in an implicit surface-atmosphere coupling (Polcher et al., 1998). Several other useful parameters are also computed, such as surface temperature ($T_S$), surface albedo and emissivity, which are needed to calculate the upwelling components of the radiative budget. The calculations are performed at the grid cell scale, by aggregating its landscape into three soil tiles: one for the forest, one for grass and crops, and one for the bare soil. The aerodynamic parameters of the grid cell correspond to the averaged parameters weighted by the effective areal fraction of each soil tile.



The raw output data of this climate simulation have been post-processed and only a variety of specific variables describing atmospheric as well as land surface conditions have been archived for further uses. These include atmospheric variables on half-eta levels ($M$) such as $q$, potential temperature ($\theta$) and horizontal wind components ($u$ and $v$). The surface data involve among others skin temperature $T_S$, $SP$, precipitation rate, $H$ and $LE$, the four components of radiative fluxes, as well as the underground liquid water content. The most conventional meteorological variables, such as $T$ and $RH$ at $2 \, \mathrm{m} \, \mathrm{agl}$ are also available. The data are stored at a temporal resolution of $3 \, \mathrm{hours}$ for all but the underground liquid water content. More specifically, the data on half-eta levels correspond to nearly instantaneous values every 3-hour starting at $00:00$ UTC. Meanwhile, the surface data, mostly provided by ORCHIDEE, consist of time-centred mean over a $3 \, \mathrm{hours}$ window and their timestamps start at $01:30$ UTC. The underground water content is archived at a daily timescale. It consists of liquid water height within different sub-layers, each sub-layer holding one node.

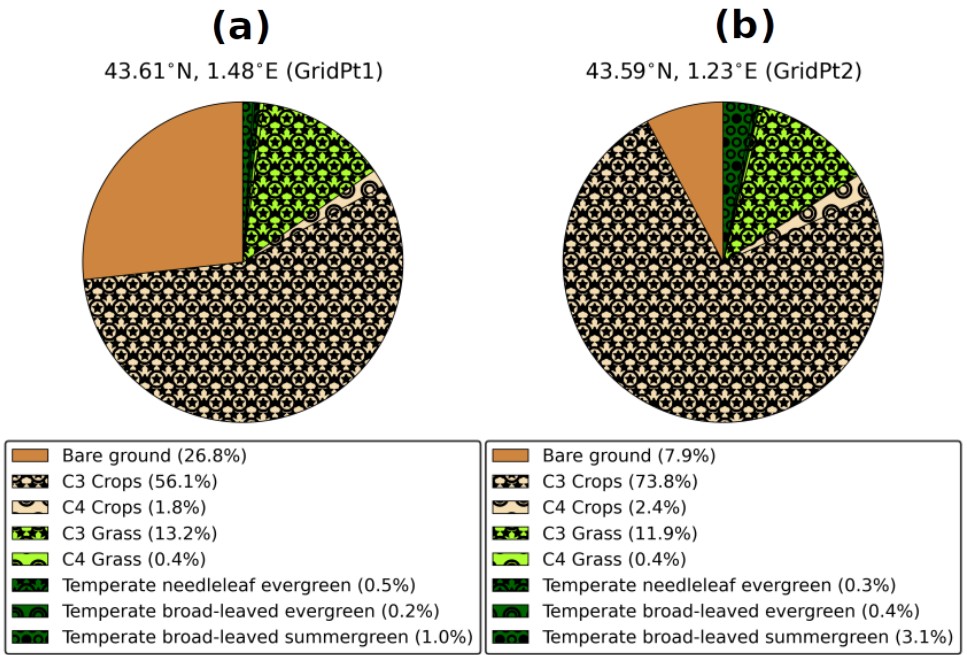

**Figure 3.** Landscape composition at the two grid cells of the RegIPSL model (GrdPt) geographically nearest to the Météopole-flux station, GrdPt1 (b) being the closest.

It would be very interesting to extract the simulation data at a grid cell with a landscape composition that resembles the
landscape contribution to the turbulent heat fluxes measured at the Météopole-flux station as found by Jomé et al. (2023). This
grid cell should also be geographically close to the station to preserve the local behaviour of atmospheric forcing. However, for
all the grid cells located within a distance $60\ \mathrm{km}$ to the station coordinates (e.g 3 times the simulation horizontal resolution),
at least $50\ \%$ of the surface is covered by crops and forest PFTs. The areal fraction of grass PFTs ranges between $10 - 21\ \%$.
The simulation data are then extracted at the two nearest grid cells to the Météopole site, as usually done. Figure 3 shows
their landscape composition. The proportion of bare soil and crops are respectively greater and smaller in the closest grid cell
(GrdPt1). Only the simulation covering the period from 01 January 2012 (the first year of heat fluxes measurements at the
Météopole station) to 31 December 2016 is considered. This period coincides with the selected observation period from 24
November 2012. The time-centred average of $q_M$, $\theta_M$, $u_M$ and $v_M$ are used for consistency with surface and soil data. The
$SWC$ at the soil nodes is calculated as the ratio of liquid water height over the thickness of the corresponding sub-layer. For
each node, the value of $SWC$ at a given day is assigned to every 3 hourly timestamp of that day. Similarly to the selected
observational data, the days with cumulative rainfall exceeding $5\ \mathrm{mm}$ were excluded, reducing the simulation data by around
$15\ \%$.

## 3.3 The data-driven statistical model

Since the fluxes $H$ and $LE$ are continuous variables, our problem is formulated as multivariate regression settings. There are
several ways to achieve regression with ANNs. The easiest is to exploit the most basic ANN, the feed-forward network also
known as the multi-layer perceptron (MLP). Because of its exceptional ability to approximate complex multivariate functions,
MLP has become the most widely used type of ANN. Accordingly, our data-driven statistical model is built using MLP.
This section begins by briefly introducing this type of ANN. Subsequently, the implementation of the statistical model with
half-hourly observational data is detailed. Finally, the challenges involved in its application to data from the climate simulation
are outlined.

### 3.3.1 The multi-layer perceptron

The elementary unit of ANNs is the mathematical neuron (Rosenblatt, 1960), which is illustrated in Figure 4. It is a numerical
computational unit that receives information via synaptic connections characterized by weights ($\boldsymbol{w}$), and provides a response



using an activation function ($f$) and a bias ($b$), as follows:

$$o = f\left(b + \sum_{j=1}^{N} w_j \cdot x_j\right) \tag{3}$$

Where $N$ is the number of input variables. Overall, the input data of $f$ and its output range within $[-1,1]$. The neuron's behaviour, either linear or non-linear, is defined by its activation function. Although there are many types of activation functions, sigmoid-like (e.g. logistic and hyperbolic tangent) and identity functions are commonly used for regression (Zhang, 2008).

The MLP is a supervised ANN, which consists of fully interconnected neurons organised in successive layers (see Figure 5 for illustration). This includes an input layer to receive the predictors, an output layer to get the outcome, and at least one intermediate layer between them, the so-called hidden layer(s). There is one neuron per input and output variable. The neurons in the input layer just carry the data without any calculations. The hidden layer(s) form the computational core of MLP. Although the topography of hidden layers (number of layers and neurons) has an impact on the network's capability to approximate the relationships, there is not yet a universal rule defining the most suited for a given problem. Thus, finding an appropriate configuration for hidden layers (number of layers and neurons) is generally a non-trivial and uphill task with expensive computational costs.

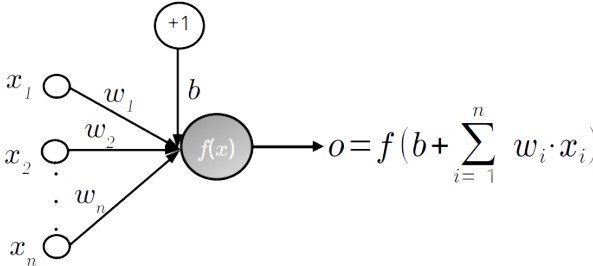

**Figure 4.** Schematic illustration of a mathematical neuron (adapted from Zhang, 2008): $x_i$, $w_i$ correspond respectively to its numerical inputs and synaptic weights whereas $o$ is the response based on its activation function ($f$) and bias ($b$). This latter is schematized by an input variable with a value and weight equal to +1 and $b$ respectively.

As a supervised ANN, MLP acquires knowledge about its task through a learning stage. During this stage, the network is provided with examples of paired predictors and desired outputs, and its weights and biases adjust accordingly. The MLP



understanding of the physics laws then entirely relies upon the quality and amount of data on which it has learnt. Thus, the
more consistent examples the more chances of the MLP being accurate over unseen input data. The learning data is usually
separated into two disjoint subsets: training and validation. The MLP weights and biases are updated using a backpropagation
optimization technique which minimizes an error metric calculated on the training data, between the MLP outputs and the
desired values. The Mean-Square-Error (MSE) is the common error metric (Zhang, 2008) for regression. In general, there are
three modes in which backpropagation optimization may be applied: (i) the 'online' mode in which the network weights and
biases are updated for each example in the training set, (ii) the 'batch' mode in which all the training data are considered at
once, and (iii) the 'mini-batch' mode which is a mixture of the two first and achieves their advantages while limiting their
inconveniences; the training data is subdivided into a smaller fixed number of samples (the mini-batch) which are used for
modifying weights and biases. The default mini-batch size is 32. Another key parameter of the learning stage is the number
of training data passages through the network, also called epochs (Chicco, 2017; Zhang, 2008; Kruse et al., 2013). Indeed,
with small epochs, the network would not understand the complexity of the data, leading to an underfitting. By contrast, too
high epochs may lead to overfitting; the network would capture all the details of the training data while performing badly on
unknown data. The validation data serve to assess the network performance during the updating of its weights and biases.
Thus, a fairly large number of epochs can be envisaged to avoid underfitting and the learning is early stopped when the
performance over the validation set no longer improves, preventing overfitting.

Thus, the implementation of MLP can be split into three main points:

- (i) Define a set of relevant predictors based on the variables to be approximated.

- (ii) Select a learning data such that it would contain sufficient examples to statistically describe the relationships between
predictors and targeted variables.

- (iii) Find suitable MLP setting (topography of hidden layer(s), activation function, etc.) through sensitivity experimen-
tation.

Before processing with MLP, data should be scaled (i) for consistency with $f$ and (ii) to circumvent the relevance of variables
due to their magnitude. Moreover, the backpropagation algorithm is stochastic, which very often leads to variability in the
final weights and biases of MLP each time the network is retrained with the same data. Indeed, the final state may correspond
to a local minimum of the error metric (Zhang, 2008). Although the difference between the MLP outputs is usually slight,
it can be a bit annoying not to get the same results. The ensemble learning approach (Ganaie et al., 2022) may be used to



limit the instability of the MLP-based estimates and to get closer to the optimal estimate. Instead of training a single MLP, this approach involves training multiple MLPs and then averaging their outputs for regression problems. One of the standard strategies for generating these MLPs is bagging (Breiman, 1996; Khwaja et al., 2015); a base MLP is trained on a redistributed

version of the original training or learning set.

### 3.3.2 Implementation

In this work, each MLP is implemented using Tensorflow-Keras (version 2.2.0, Abadi et al., 2016), a Python library specifically designed for ANNs, known for its user-friendly interface. Unless otherwise mentioned, the default parameters are used. The

290 three points mentioned above are addressed as follows:

– (i) As MLP predictors, we looked for standardized variables that can be derived from both observational and modelled data while still having the same physical meaning. Theoretically, $H$ and $LE$ are quasi-constant within the surface layer and strongly related to near-surface radiative, thermodynamic and dynamical forcing. Moreover, their relative predominance is controlled by the wetness of the uppermost part of the soil. In numerical simulations, the atmospheric level just

295 above the ground is usually considered to be the top of the surface layer. Based on the observed variables in Table 1 and the simulation variables that have been archived, we derive a set of 9 physical variables that may analogously describe the environmental forcing in the surface layer. These include the total energy governing the surface processes ($R_{net}$), the meteorological conditions in the surface layer ($\theta_{sl}$, $\Delta_\theta$, $q_{sl}$, $\Delta_q$, $u_{sl}$, $v_{sl}$, $\Delta_U$) and the moisture in the uppermost soil layer ($SM$). Eventually, 4 trigonometric temporal coordinates are added for the description of seasonal ($d_x$, $d_y$) and

300 diurnal ($h_x$, $h_y$) cycles. The formulations of all these 13 variables, used as MLP predictors, are presented in Table 2. Under the observed environment, $SM$ is defined by $SWC_{-10cm}$ (the first depth of soil moisture measurements) and the meteorological variables are calculated assuming that the surface layer always extends above 10 magl. Under the simulated environment, $SM$ then corresponds to $SWC_{-12.3cm}$ (the nearest node to the measurement depth). Moreover, the meteorological variables are directly taken at the first half-eta level (M=1, around 8 magl) instead of diagnostic variables

as much as possible. In both environments, $\Delta_U$ is calculated assuming a null horizontal wind speed at the ground. Due to the missing data, the sunrise hour required to compute $h_x$ and $h_y$ was retrieved using the astral package (version 3.0), and rounded to the nearest half hour.



– (ii) To train the statistical model over as many multivariate cases as possible, the observational data from the five most covered years (Figure 2) were gathered as the learning set. The remaining data was reserved for assessing its ability to generalize, it will be referred to as the test set.

– (iii) Since $H$ and $LE$ are complementary fluxes and to avoid excessive sensitivity experiments in the search for a relevant architecture for MLP, the neurons in the output layer were set to 2, one for each flux. Following the literature (Kumar et al., 2011; Leufen and Schädler, 2019; Kelley and Pardyjak, 2019; Kelley et al., 2020), hyperbolic tangent and identical functions were used as the activation functions for neurons of the hidden and output layers, respectively. The weights and biases optimization was carried out with the Adam-amsgrad backpropagation algorithm (Kingma and Ba, 2017; Reddi et al., 2018) in a default mini-batch mode, and MSE was chosen as the error to minimize. Based on preliminary results, we opted for the same strategy of network training used by Leufen and Schädler (2019). The network update stops at most after 1000 epochs. Otherwise, the update is stopped early if the MSE on the validation data does not improve after 50 successive epochs, and then the network state is finally set to the epoch with the best MSE.

The input and output data are scaled to the interval [0, 1] similarly to Leufen and Schädler (2019) as follows:

$$\tilde{x} = \frac{x - x_{min}}{x_{max} - x_{min}} \tag{4}$$

Where $x_{min}$ and $x_{max}$ correspond to the extreme values of variable $x$. They are set to the theoretical values of the trigonometric functions (e.g. $-1$ and $1$ respectively) for the four temporal coordinates. For the other nine physical parameters, these values are set such that the resulting interval strictly holds both observational and RegIPSL data (see Figure A1).

The MLP architecture used in this study includes two hidden layers with $4$ and $3$ neurons respectively (Figure 5). It was found through several series of sensitivity experiments using the learning data (not shown). A bagging-based strategy was used to incorporate the inter-annual variability of $H$ and $LE$ when building the statistical model. Indeed, a $5$-fold cross-validation (Andersen and Martinez, 1999) with year-wise data splitting was applied to the learning data to generate $55$ bagged MLPs, such that the statistically-based fluxes are the average across the individual MLP outputs. Cyclically, the data from one year were used as the validation data whilst the others composed the training data, and, 11 MLPs were trained by randomly initializing weights and biases along with a shuffling of the training data before composing mini-batch subsets. Thus, each example in the learning data was at least once used as validation or training data. Although this number of 11 was





**Table 2.** MLP input variables derived from observational data and their equivalent extracted from RegIPSL data. $dd$, $\Delta h$ and $N_y$ included in the expressions of the temporal coordinates, respectively stand for Julian date, hours relative to sunrise on $dd$ and number of days in the year.

| Observations | RegIPSL |
|---|---|
| Radiative forcing at the surface | |
| $R_{net}$ | $R_{net}$ |
| Thermodynamic and dynamic in the surface layer | |
| • $\theta_{sl} = \dfrac{\theta_{10m} + \theta_{2m}}{2}$ ; $\Delta_\theta = \left.\dfrac{\Delta\theta}{\Delta z}\right\vert_{2m}^{10m}$ <br><br> • $q_{sl} = \dfrac{q_{10m} + q_{2m}}{2}$ ; $\Delta_q = \left.\dfrac{\Delta q}{\Delta z}\right\vert_{2m}^{10m}$ <br><br> • $[u,v]_{sl} = [u,v]_{3.7m}$ ; $\Delta_U = \left.\dfrac{\Delta U}{\Delta z}\right\vert_{surface}^{3.7m}$ | • $\theta_{sl} = \theta_{M=1}$ ; $\Delta_\theta = \left.\dfrac{\Delta\theta}{\Delta z}\right\vert_{surface}^{M=1}$ <br><br> • $q_{sl} = q_{M=1}$ ; $\Delta_q = \left.\dfrac{\Delta q}{\Delta z}\right\vert_{2m}^{M=1}$ <br><br> • $[u,v]_{sl} = [u,v]_{M=1}$ ; $\Delta_U = \left.\dfrac{\Delta U}{\Delta z}\right\vert_{surface}^{M=1}$ |
| Underlying soil wetness | |
| $SM = SWC_{-10cm}$ | $SM = SWC_{-12.3cm}$ |
| Temporal coordinates | |
| • $d_x = cos(2\pi \cdot \frac{dd}{N_y})$; $d_y = sin(2\pi \cdot \frac{dd}{N_y})$ <br><br> • $h_x = cos(2\pi \cdot \frac{\Delta h}{24})$; $h_y = sin(2\pi \cdot \frac{\Delta h}{24})$ | • $d_x$ ; $d_y$ <br><br> • $h_x$ ; $h_y$ |

arbitrarily chosen following the available computing resources, it ensures the repeatability of the estimates.


### 3.3.3 Application to data from numerical model

The key objective of our evaluation method is to approximate turbulent heat fluxes that may be observed under modelled environmental forcing using an MLP-based statistical model built with observational data. This implicitly assumes that the

modelled and observed data share a common input space and that the learning data is representative of this space. However, if the model data do not have similar structures (distribution and interval ranges of input variables) to the observations, the statistical model may poorly perform. Indeed, MLP has a good interpolation capability but may not extrapolate well beyond the ranges of values it has learnt on (Bonnasse-Gahot, 2022). For four of the nine physical variables ($\Delta_\theta$, $\Delta_q$, $u_{sl}$ and $v_{sl}$) RegIPSL data spread beyond the observed values (Figure A1). A rigorous application of machine learning techniques typically



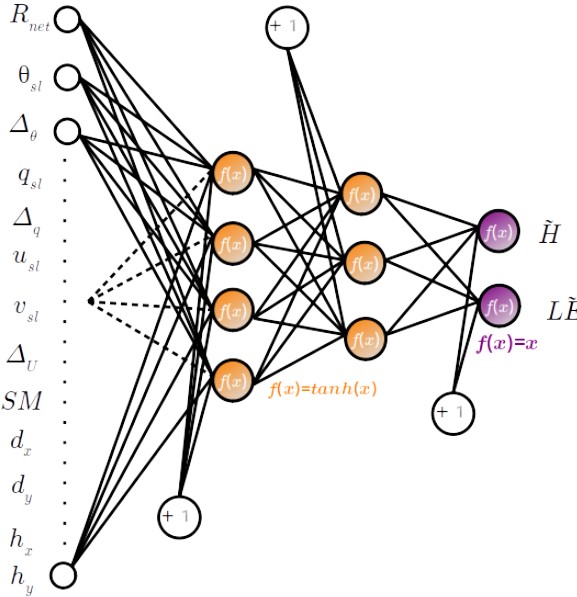

**Figure 5.** Architecture of MLP used in this study. The input variables are described in table 2.

requires the use of transfer learning to mitigate the loss of performance when trained ANN is applied to data originating from another source (Day and Khoshgoftaar, 2017). Since the observed $H$ and $LE$ associated with the modelled environment are unknown, the most challenging transfer learning approach, unsupervised domain adaptation, would normally be used in our case. Numerous methods are available for achieving unsupervised domain adaptation. We tried the easiest and most popular methods over the RegIPSL input data such that correlation alignment (Sun et al., 2015), feature augmentation (Daumé III,

2009), subspace alignment (Fernando et al., 2013), transfer component analysis (Pan et al., 2011) and feature selection (Uguroglu and Carbonell, 2011) as implemented in ADAPT library (version 0.4.2, de Mathelin et al., 2023). Either, we get unreasonable fluxes, particularly in stable conditions, or the fluxes vary from one method to another, so it is hard to conclude the most efficient. The most sophisticated methods require the use of an encoder which may be an ANN, with the laborious and time-consuming task to find its adequate configuration. Further investigation is needed to find a consistent unsupervised

domain adaptation method, but that is beyond the scope of this paper.

Finally, our proposed evaluation method does not currently include any transfer learning method. Under the traditional assumption that training and testing data come from the same distribution and input space (Aggarwal, 2014), the MLP-based statistical model is directly applied to RegIPSL input data. To get an insight into the loss of performance, the relative

contribution of predictors towards the fluxes estimates is calculated using the SHapley Additive exPlanations (SHAP) algo-



rithm (Lundberg and Lee, 2017). SHAP is attractive because it unifies several common methods for interpreting approximation with ANN. It is based on the game theory approach; for an individual game (MLP outputs), contributions (called SHAP values) are assigned to each player (predictor). The mean of SHAP absolute value across several instances is then used to measure the predictor influence. The higher the corresponding SHAP value, the more contributing the input variable on average.


## 4 Assessment of the statistical model

As mentioned above, the data-driven model consists of 55 MLPs that were trained under a 5-fold cross-validation applied to a collection of 5 years of observational data. The estimated fluxes are then the average of their outputs. The most important goal in machine learning is generalization, e.g. the data-driven model should perform well not only on learning data but also on

unseen data. This section discusses the performance of our statistical model on learning and test sets.

Figure 6 compares half-hourly MLP-based fluxes against their targeted values for learning and test sets separately (left and right panels respectively). It shows $H$ and $LE$ (top and middle panels respectively) and the total turbulent heat flux $H + LE$

(bottom panels). Overall, the root-mean-square-error (RMSE) range between $20 - 30$ $\mathrm{Wm}^{-2}$ and the Pearson's correlation coefficient (r) are greater than $0.95$, indicating a very good agreement between estimated and observed fluxes. Interestingly, the total turbulent heat flux is particularly well approximated, despite not being a direct output of MLPs. Furthermore, the RMSE increases by less than $3\%$ and the correlation is almost the same from learning to test sets. Thus, on both known learning data and unknown test data, the statistical model performs quite similarly on average, demonstrating that it generalizes rather well.

Besides this remarkable performance, there are some noticeable shortcomings shown in both learning and test sets. On the one hand, the estimated $H$ is limited to around $-50$ $\mathrm{Wm}^{-2}$ while values of less than $-100$ $\mathrm{Wm}^{-2}$ are observed (Figure 6a and b). On the other hand, the observed $LE$ greater than $200$ $\mathrm{Wm}^{-2}$ tends to be underestimated (Figure 6c and d), especially for the test set. Thus, the statistical model has issues with estimating large $LE$ and negative $H$ associated with a very unstable and stable surface layer respectively.




**Figure 6.** Half-hourly statistical estimates (MLP-based) of sensible heat flux ($H$, a and b), latent heat flux ($LE$, c and d) and total turbulent heat flux ($H + LE$, e and f) with observed input data against observed fluxes, for learning (a, c and e) and test (b, d and f) sets. The values at the top of each panel correspond to Root-Mean-Square Error (RMSE) and Pearson's correlation coefficient. The lines in magenta and orange represent the linear regression and identical fits respectively.

Figure 7 shows the cumulative distribution functions (CDFs) of RES (eq. 2) calculated with half-hourly observed $R_{net} - G$ together with observed and MLP-based fluxes. For both learning and test sets (Figure 7 a and b resp.), CDFs of observed

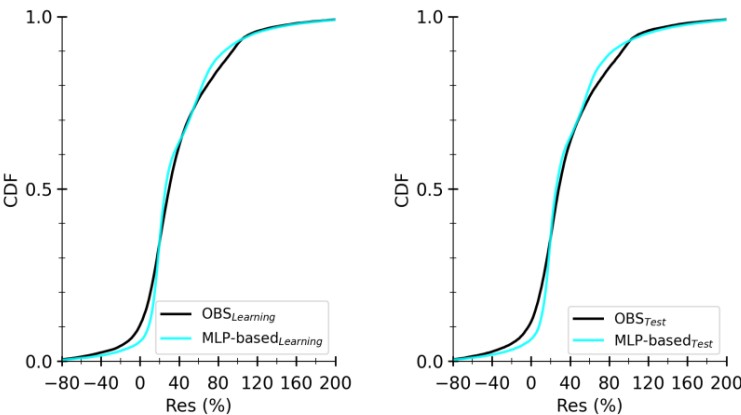

**Figure 7.** Cumulative density functions (CDFs) of RES (eq. 2) calculated with half-hourly observed turbulent heat fluxes (black) and competing MLP-based estimates with observed input data (cyan) for learning set (a) and test set (b).

fluxes and estimated fluxes nearly coincide. Although the sample size is different for these two sets, their CDF curves look

quite similar, implying that they are individually representative of the main local characteristics of energy imbalance and

that the statistical model reproduces quite well. Closer inspection showed that the statistical model produces smoothed fluxes

that preserve the observed relationship between $H + LE$ and $R_{net} - G$ (not shown). This smoothing is the main cause of

intermittent departures between the CDF curves. Overall, the CDFs are smaller than $0.25$ for RES lower than $20\ \%$ indicating

that both observed and estimated $H + LE$ are smaller than $R_{net} - G$ in most of the cases. This tendency is systematic for large

$H + LE$ (not shown). Thus, the statistical model carries the limitations of observed turbulent heat fluxes. The representativity

of local measurements of turbulent heat fluxes in heterogeneous landscapes with the EC technique is being investigated as part

of the MOSAI project (e.g. Jomé et al., 2023).

Table 3 compares RMSE, r and coefficients of the linear regression (slope and intercept) when the statistical model is applied

to the half-hourly raw observational data (in Figure 6) and 3-hourly averaged values similarly to RegIPSL data. It provides

insights into the performance at the temporal resolution of simulation data. The values of these metrics show that the statistical

model performs better at the temporal resolution of $3\ \mathrm{hours}$.

Figure 8 shows the composite seasonal cycles of observed and MLP-based turbulent heat fluxes for the learning and test sets

(left and right panels respectively). The observed $H + LE$ presents quite similar seasonal cycles for both sets, with a peak

from April to September (Figure 8c and f). However, observed $H$ and $LE$ do not typically present the same seasonal cycle.

Indeed, from April to September, $H$ ($LE$) is on average stronger (weaker) in the learning set. In this set, $LE$ predominates





**Table 3.** Comparison of Root-Mean-Square-Error, linear correlation and the linear regression fitting coefficients (slope and intercept) when applying the MPL-based statistical model to half-hourly raw and to 3-hourly average observational data. The statistical model has been constructed using the half-hourly version of the learning set (Figure 2a).

|  |  | Learning set | | | Test set | | |
|---|---|---|---|---|---|---|---|
|  |  | $H$ | $LE$ | $H+LE$ | $H$ | $LE$ | $H+LE$ |
| RMSE $(\mathrm{Wm}^{-2})$ | 30 mn | 21.9 | 20.4 | 25.9 | 25.2 | 24.2 | 30.8 |
|  | 03 hr | 17.6 | 16.5 | 16.1 | 20.7 | 19.3 | 20.7 |
| Correlation | 30 mn | 0.98 | 0.97 | 0.99 | 0.96 | 0.95 | 0.98 |
|  | 03 hr | 0.98 | 0.97 | 0.99 | 0.97 | 0.97 | 0.99 |
| Slope | 30 mn | 0.94 | 0.91 | 0.97 | 0.98 | 0.89 | 0.96 |
|  | 03 hr | 0.96 | 0.93 | 0.99 | 1.01 | 0.91 | 0.98 |
| Intercept | 30 mn | 3.56 | 5.46 | 4.22 | 5.77 | 1.29 | 3.43 |
|  | 03 hr | 1.44 | 4.96 | 1.51 | 3.50 | 0.70 | 0.38 |

over $H$ from March to June and it is the reverse in the subsequent months until October (Figure 8a and b). In the test set, the predominance of $H$ starts later in August, since $LE$ slowly decreases from May to July (Figure 8d and e). This highlights the inter-annual variability in the partitioning of turbulent fluxes across the period from late spring to late summer, as shown in Figure A2, that different vegetation dynamics may explain.

Overall, the MLP-based fluxes correctly reproduce the observed signal of seasonal cycles along with day-to-day variability for both learning and test sets. This is particularly true for the total turbulent heat flux (Figure 8c and f). Interestingly, the relative predominance between $H$ and $LE$ in the two sets is remarkably well replicated. In the learning set, the absolute difference between estimated and observed fluxes remains smaller than $5~\mathrm{Wm}^{-2}$ for all the fluxes. However, in the test set, striking differences appear between May and September, when the total turbulent heat flux is at its maximum. Indeed, while the MLP-based $H+LE$ is quite similar to observations (Figure 8f), the MLP-based $H$ overestimates observations (Figure 8d), by more than $10~\mathrm{Wm}^{-2}$ in June and July, and, the MLP-based $LE$ underestimates observations, by more than $14~\mathrm{Wm}^{-2}$ from July to September (Figure 8e). This comes from the fact that the MLPs have learnt on respectively weaker $LE$ and stronger $H$ on average (Figures 8 and A2).

In conclusion, the statistical model, constructed with half-hourly observational data, provides highly consistent estimates of $H$ and $LE$. It especially approximates very well the total turbulent heat flux $H+LE$. Its performance in terms of RMSE and linear correlation ranges within the best reported by the literature on surface parametrization schemes (e.g. Liu et al., 2013;





**Figure 8.** Composite monthly average of sensible heat flux ($H$, a and d), latent heat flux ($LE$, b and e) and total turbulent heat flux ($H + LE$, c and f) observed and statistically estimated with input observed data (lines in black and cyan respectively) for the learning and test sets (left and right panels respectively). The solid lines represent the means and the error bars correspond to 10 and $90^{th}$ percentiles, calculated by gathering the daily averages of half-hourly samples.

Leufen and Schädler, 2019; Román-Cascón et al., 2021). The non-closure of SEB embedded in the observed fluxes is also replicated. The performance improves considerably at 3-hour and seasonal timescales, probably due to a reduction of noise in the observational data. In particular, the relative predominance between heating and evaporation is faithfully reproduced. All these results underlie that the statistical model captured rather well the fundamental links between turbulent heat fluxes and environmental factors. However, the statistical model shows limitations in estimating a very large $LE$ and negative $H$, notably at hourly timescale. Moreover, it struggles to generalize the magnitude of $H$ and $LE$ in the spring and summer seasons. Indeed, the limited number of examples in the learning set does not cover all the inter-annual variability.

Increasing the learning data at the expense of the test data would no doubt improve the skill of the statistical model on observed

fluxes. However, uncertainties will always remain when it is applied to unseen data and cannot be assessed in the absence of

corresponding observations, as is the case for numerical models. By convention, thirty years of observational data are required

for good climatological coverage while around ten years is currently available at the Météopole-flux station. The spring and

summer seasons are usually characterized by intense vegetation activity. Furthermore, the relationship between near-surface

latent heat flux and soil moisture could be modulated by the vegetation state (e.g. Román-Cascón et al., 2020). Yet, the MLP

input variables lack one variable that can characterize the dynamics of the vegetation, such as the leaf area index (LAI). Such

a parameter will then contribute to a more accurate description of the inter-annual variability of the near-surface turbulent heat

fluxes. Hence, its use would probably improve the generalization ability of the statistical model.

## 445  5  Evaluation of simulated near-surface turbulent heat fluxes using the statistical model

In this section, the near-surface turbulent heat fluxes simulated by the RegIPSL model, from 01 January 2012 to 31 December

2016, are evaluated by focusing on the weaknesses of its land surface scheme. The benefits and limitations of our proposed

evaluation approach are emphasized.


Figure 9 presents the means of SHAP absolute value for each input variable, averaged across the trained MLPs and the

estimates under observed (learning and test sets) and simulated (at the two nearest grid cells) environments. The variables are

ranked on the $y$-axis in descending order according to their relevance at GrdPt1, the closest geographical grid cell. Note that,

the SHAP value increases with the variable relative contribution. Thus, the possible loss of performance when the input data

ranges beyond the learning interval could be discussed.

In both the observed and simulated environments, $R_{net}$ is by far the major contributing variable. $\theta_{sl}$ is the third most

contributing physical variable in both $H$ and $LE$ estimates, regardless of the environment. The importance of the other

inputs varies with the environment, the grid cells and the fluxes. Nonetheless, $SM$ is always among the four leading physical

variables. Thus, the trained MLPs composing the statistical model clearly understood that the net radiative budget at the

surface is the primary driver of turbulent heat fluxes and that the soil wetness is a crucial factor for the partitioning between





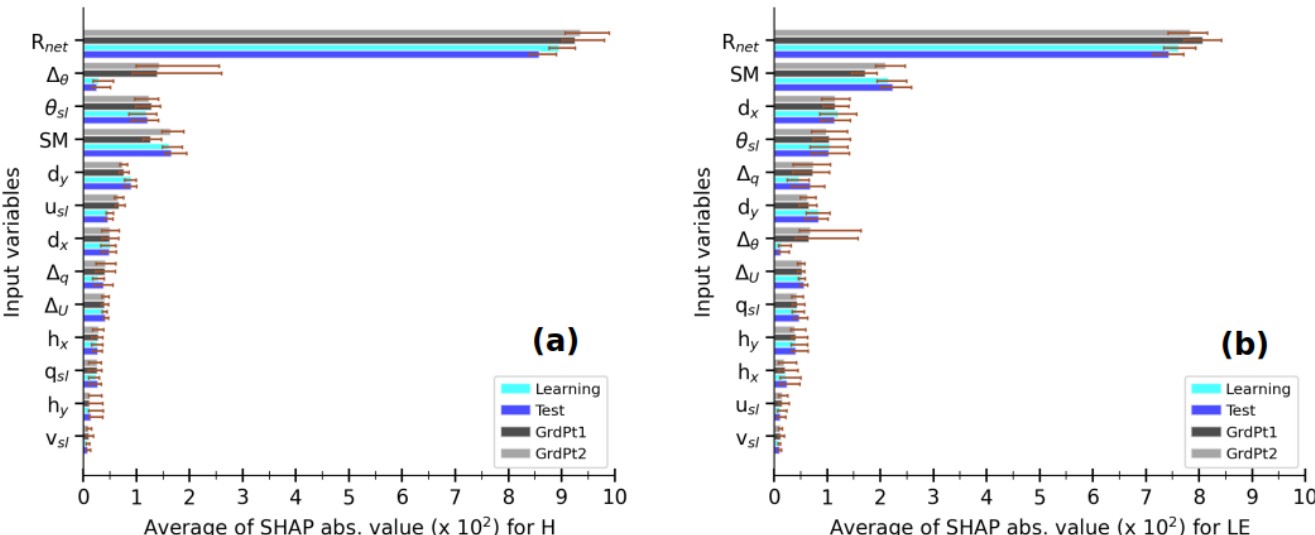

**Figure 9.** Averages of SHAP absolute value for each input variable in statistical estimates of sensible heat flux ($H$, a) and latent heat flux ($LE$, b) under the simulated environment at GrdPt1 and GrdPt2 and, under observed learning and test sets, according to the caption. For a given MLP, the SHAP absolute values are calculated for each estimate and then averaged over the samples of each data set. The coloured bars indicate the median values and the error bars correspond to the $10^{th}$ and $90^{th}$ percentiles across the 55 trained MLPs that compose the statistical model. In each panel, the input variables are ranked in descending order under the environment at GrdPt1.

heating and evaporation. The contribution of $\Delta_\theta$ in the simulated environment considerably varies with the trained MLPs. Moreover, it is the second most contributing variable for $H$ estimates in this environment. Whereas, $\Delta_\theta$ is one of the less influential variables in the observed conditions. This demonstrates that the statistical model considers the change in the

environmental context. Under the simulated conditions, the contribution of $\Delta_\theta$ is quite comparable to that of $SM$ and $\theta_{sl}$. Thus, the aggregated contribution of the three physical variables ($\Delta_\theta$, $u_{sl}$ and $v_{sl}$) whose simulated values spread beyond the learning interval is overall smaller than 20 % of the aggregated contribution of $R_{net}$, $SM$ and $\theta_{sl}$. Therefore, we hypothesise a minor loss of performance due to extrapolation when the statistical model is directly applied to data from the RegIPSL model.

The scatter plots in Figure 10 illustrate the consistency of the MLP-based fluxes in the simulated environment as well as the errors due to different environmental forcing. It compares 3-hourly MLP-based fluxes with environmental conditions at GrdPt1 against the corresponding 3-hourly time-centred average of observed fluxes (left panels) and MLP-based fluxes in corresponding observed conditions (right panels). It only includes the timestamps between 24 November 2012 to 31 December 2016 for which both modelled and corresponding observational data are available, accounting for around 40 % of the model

sample data. In each panel, the flux varies within the same interval range, between $-50$ and $400$ $\mathrm{Wm^{-2}}$ for $H$ and $LE$, and

between $-50$ and $500$ Wm$^{-2}$ for $H + LE$. Moreover, the correlation coefficient is around $0.9$ or greater, indicating that the variability in MLP-based fluxes with data from the RegIPSL model is consistent with actual observations (EC measurements or MLP-based estimates). Hence, the differences between the fluxes mostly lie in their magnitudes. These findings also hold for the second grid cell (Figure A3). Since the difference is much more pronounced for large fluxes, the divergence would

occur mainly during daylight hours. The MLP-based $H$ and $H + LE$ associated with the simulated environment are stronger than observed. It conforms to the tendency found when comparing simulated $R_{net}$ to observed (not shown).

Figure 11 compares 3-hourly simulated turbulent heat fluxes at GrdPt1 to observations (left panels) and statistical estimates

under simulated environment (right panels) at the same timestamps as in Figure 10. The scatter is considerably reduced with a better alignment along the linear regression fit when the MLP-based fluxes are instead used as the reference values. These changes are in agreement with a reduction of uncertainties, particularly those related to the difference in environmental conditions. There is no ideal fit between simulated $H + LE$ and corresponding MLP-based estimates (Figure 11f) because the statistical model does not allow SEB closure (Figure 7) and an important part of $R_{net}$ is converted to soil heat flux throughout

the simulation (not shown). This indicates that some substantial external uncertainties remain. Nonetheless, comparing with MLP-based estimates highlights the shortcomings of the surface scheme better than comparing with observed data. According to the MLP-based fluxes in the simulated environment, the surface scheme tends to quasi-systematically underestimate large $H$ (Figure 11a and b). This tendency is more pronounced for the second grid cell (Figure A4), which comprises a greater fraction of crops. Whereas, the simulated environment promotes overestimating large observed $H$ (Figure 10a and b). Thus, the bias in

simulated fluxes may be due to i) incorrect surface heterogeneities (with crop instead of grass and bare soil instead of urban area) together with ii) inadequate formulations of fluxes.

Furthermore, the simulated fluxes at the two grid cells differ very slightly. As a result, the comparison with observations leads to nearly similar RMSE and correlation. Meanwhile, the use of MLP-based estimates shows better performance at GrdPt1, since the landscape at GrdPt2 induces drier soil on average (Figure A1i). Thus, it appears that the $H$ and $LE$ formulations in

ORCHIDEE lack sensitivity to soil wetness.

Another key advantage of our method is that all the simulation timestamps can be used since the availability of the same timestamps in the observations is no longer needed. The statistical significance of the comparison is thus enhanced. Overall,





**Figure 10.** 3-hourly MLP-based fluxes in the simulated environment at GrdPt1 against observations (a, c and e) and MLP-based fluxes in the observed environment (b, d and f) at Météopole. It only considers model 3-hourly timestamps between 24 November 2012 and 31 December 2016, for which observational data are available. The values at the top of each panel correspond to the number of samples ($N$), Root-Mean-Square-Error ($RMSE$) and Pearson's correlation coefficient ($r$). The lines in magenta and orange represent the linear and identical fits respectively.

including all the timestamps between 01 January 2012 and 31 December 2016 in the comparison does not change a lot the







**Figure 11.** 3-hourly simulated fluxes at GrdPt1, sensible heat flux ($H$, top panels), latent heat flux ($LE$, middle panels) and total turbulent heat flux ($[H + LE]$, bottom panels) against observations (a, c and e) and MLP-based fluxes under simulated environment (b, d and f). The same timestamps as in Figure 10 are also considered here. The values at the top of each panel correspond to the number of instances ($N$), Root-Mean-Square-Error ($RMSE$) and Pearson's correlation coefficient ($r$). The lines in magenta and orange represent the linear and ideal fits respectively.

previous finding that the surface scheme struggles with large $H$ (Figure A5).





In Figure 12, the comparison of seasonal cycles of $H$, $LE$ and $H + LE$ between simulation (magenta), observations (black), MLP-based estimates in the observed environment (cyan) and MLP-based estimates in simulated environment (orange) shows that:

– The total flux $H + LE$ is strongly affected by the environmental conditions, and the non-closure problem does not seem to influence a lot the results of comparison with both observations and MLP-based estimates. Indeed, the data-driven model captures the non-closure of SEB in the observations (section 4, Figure 2). Hence, if this effect was so strong the MLP-based $H + LE$ in the simulated environment would have been much weaker than the simulated $H + LE$, which is not necessarily the case. Thus, the differences between observations and simulations mainly rely on the overestimation

of $R_{net}$ in the simulation during the spring and summer months (From April to August, Figure A6). This overestimation of $R_{net}$ is likely due to more shortwave radiation reaching the surface because of a lack of clouds, as already mentioned by several modelling studies over mid-latitude (e.g. Cheruy et al., 2014; Bastin et al., 2018; Chakroun et al., 2018).

    – Strikingly, the partitioning between $H$ and $LE$ in June, July and August differs between simulated fluxes and MLP-based estimates in the simulated environment. The effective fraction occupied by the crops is at its maximum during

the summer months (not shown). The larger $R_{net}$ in the simulation mainly leads to higher simulated $LE$ than observed. Whereas, this stronger energy is converted into higher $H$ and weaker $LE$ by the statistical model, especially at the grid cell with the smaller fraction of bare soil. This is consistent with the fact that urban surfaces are not represented in ORCHIDEE and are replaced by bare soil, which evaporates more than impervious surfaces.

    – As mentioned above, the simulated heat fluxes are not very well sensitive to soil wetness whilst the MLP-based estimates

are. Hence, the two grid cells show important differences for the MLP-based fluxes while not for the simulated fluxes. The variable $SM$ is weaker at GridPt2 than at GridPt1 between May and September (not shown), which explains higher MLP-based $H$ and lower MLP-based $LE$ during this period. The same results are found when all the diurnal cycles of the model sample data are considered (Figure A7). This deficiency opens an avenue for improvements in ORCHIDEE.


There is clear evidence that the simulated $H$ and $LE$ are highly biased from late spring to late summer. This is undoubtedly due to inappropriate representation of the land cover together with inaccurate weather conditions. Our statistical model also shows low generalization ability over this period (Figure 8), illustrating the challenge of using environmental variables to parameterize surface turbulent heat fluxes that include contributions from heterogeneous patches. Nonetheless, the use




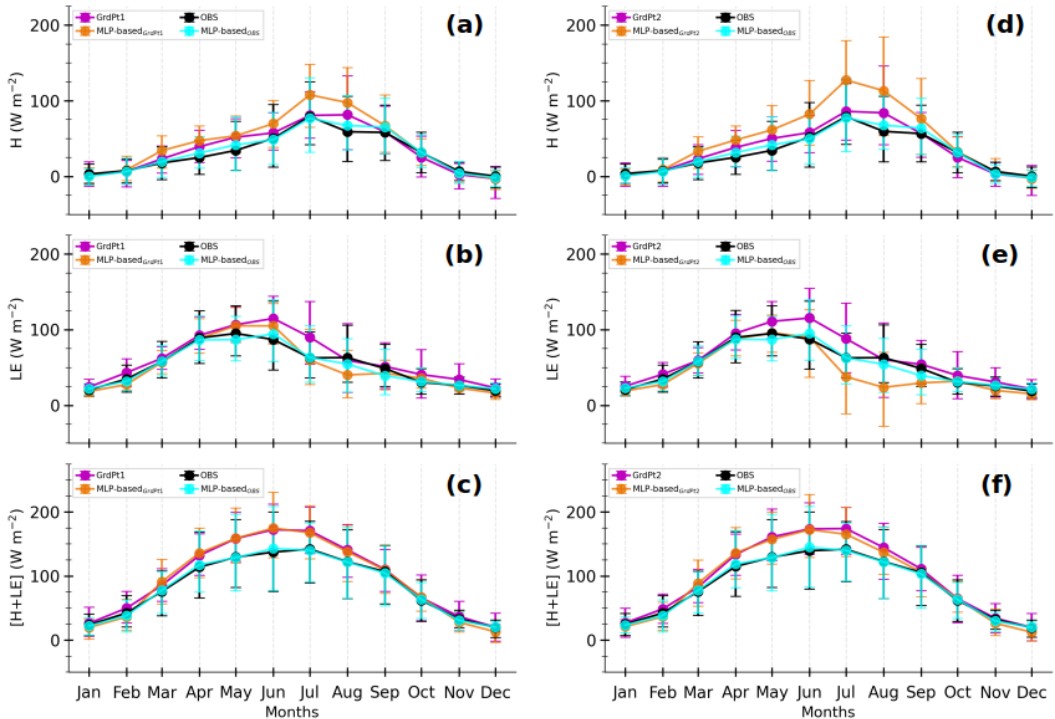

**Figure 12.** Composites monthly averages of simulated sensible heat flux ($H$, a and d), latent heat flux ($LE$, b and e) and total turbulent heat flux ($H + LE$, c and f) at GrdPt1 and GrdPt2 (magenta lines in left and right panels respectively) and that of their respective observations at the Météopole (black lines) and MLP-based estimates in observed and simulated environments (cyan and orange lines respectively). The solid lines correspond to the means and the error bars represent the 10 and $90^{th}$ percentiles, calculated by gathering the daily averages using 3-hourly data. Only, the timestamps between 01 December 2012 and 31 December 2016 common to both simulation and observational data have been considered. The days involving less than 6 timestamps were excluded.

of this statistical model shows very consistent differences between the two grid cells that are not so evident when using observations. Indeed, the MLPs were trained on observed fluxes involving urban contribution, and urban surfaces were replaced by bare soil in the version of ORCHIDEE used in the RegIPSL model. Yet, the bare soil evaporates much more and heats less than impervious surfaces such as urban areas. As expected, the errors are lower at the grid cell with a greater fraction of bare soil. According to the MLP-based fluxes, the environmental forcing discrepancies between observations

and simulations appear to be the main source of uncertainty for a direct comparison at the seasonal scale, compared to the recurrent SEB imbalance in observations. Within the simulated environment, the surface scheme in the RegIPSL model tends to underestimate large $H$ and then inversely overestimate the associated $LE$. This certainly contributes to a poor representation of intense convective heating of the lower atmosphere during the late summer when heating predominates over evaporation.



## 6   Conclusions


The representation of the near-surface turbulent heat fluxes $H$ and $LE$ is the second most important source of error in the numerical weather and climate simulations. However, it is very challenging to unambiguously quantify this error with the existing evaluation methods. In the framework of the MOSAI project (Lohou et al., 2022), this study proposes a different evaluation approach when a long period of comprehensive observational data is available. Based on the observed cases, a

data-driven statistical model is first developed to approximate observed $H$ and $LE$ with near-surface environmental factors as inputs. The data-driven model is then applied to the simulated environment to generate potentially observed fluxes under this environment. By comparing the simulated fluxes against the statistically-based estimates, the evaluation is performed in the environment as seen by the numerical model.

A demonstration study was carried out with about 10 consecutive years of observational data acquired at one of the permanent French instrumented sites of the ACTRIS research network. The data-driven statistical model is a collection of several multi-layer perceptrons, trained on the data of the 5 most covered years after cleaning. Thirteen variables characterizing the environmental forcing in the surface layer are used as inputs to simultaneously provide estimates of $H$ and $LE$. The analysis of variables contribution showed that the estimates are largely based on three classical physical parameters, namely

the surface net radiative flux, the mean potential temperature of the surface layer and the wetness of underlying soil. This opens the possibility to reduce the number of input parameters. Overall, the estimated fluxes under observed conditions are rather consistent with the observed fluxes for known and unknown cases by the MLPs. Similar to the observed fluxes, the estimated fluxes do not close the Surface-Energy-Budget. Thus, this data-driven model can be convenient for gap-filling, especially under wet conditions. Nevertheless, it does not correctly approximate negative $H$ and tends to underestimate large

$LE$. Moreover, its ability to generalize is altered from spring to late summer, likely because the leading input parameters do not fully describe the strong inter-annual variability in this period. This limitation can probably be overcome by adding a typical vegetation parameter (e.g. $LAI$) to the inputs.

The data-driven model was subsequently applied to a regional climate simulation performed with the RegIPSL model to freeze

the uncertainties which may come from the inaccuracy of simulated environmental forcing. The simulation data were extracted at the two nearest grid cells to the station. The comparison between simulated and observed fluxes gives the error resulting from the compensation between the components of the numerical model. The noticeable difference is found from late spring to late summer, in agreement with the previous studies. Overall, both simulated $H$ and $LE$ are stronger than observed, in

consistency with stronger net radiative flux. The comparison of simulated and statistically-based heat fluxes in the simulated

environment revealed that the incorrect surface characteristics in the grid cells mainly cause an underestimation of large $H$.
Moreover, the partitioning between heating and evaporation is not very sensitive to soil moisture.

By circumventing the challenge of comparing the turbulent heat fluxes from different environments, our evaluation method
offers promising perspectives for adequate evaluation of the surface parametrization schemes. The ACTRIS France network

offers the possibility of applying this methodology to other supersites where the variables required for this analysis have
been also measured for several years, allowing investigation in different types of surfaces and climates. Thanks to the
MOSAI project, each of these sites has benefited from a one-year enhanced observing period for a better characterization
of the representativity of the long-term fluxes stations, which is crucial information for model evaluation. Moreover, the
statistical model could be developed under the fundamental assumption of SEB closure (e.g., Hu et al., 2021) used in the

numerical weather and climate models. There is also a need to include a transfer learning strategy to prevent the possible
loss of performance when the statistical model is applied to cases with the leading input parameters ranging out of its known
domain. Besides, this approach could be used to evaluate community numerical simulations like reanalysis and to revisit the
intercomparison of land-surface schemes.

*Data availability.* The data from the Météopole are freely available on AERIS platform (https://www.aeris-data.fr/). The source for RegIPSL
data is indicated in the acknowledgements.




## Appendix A

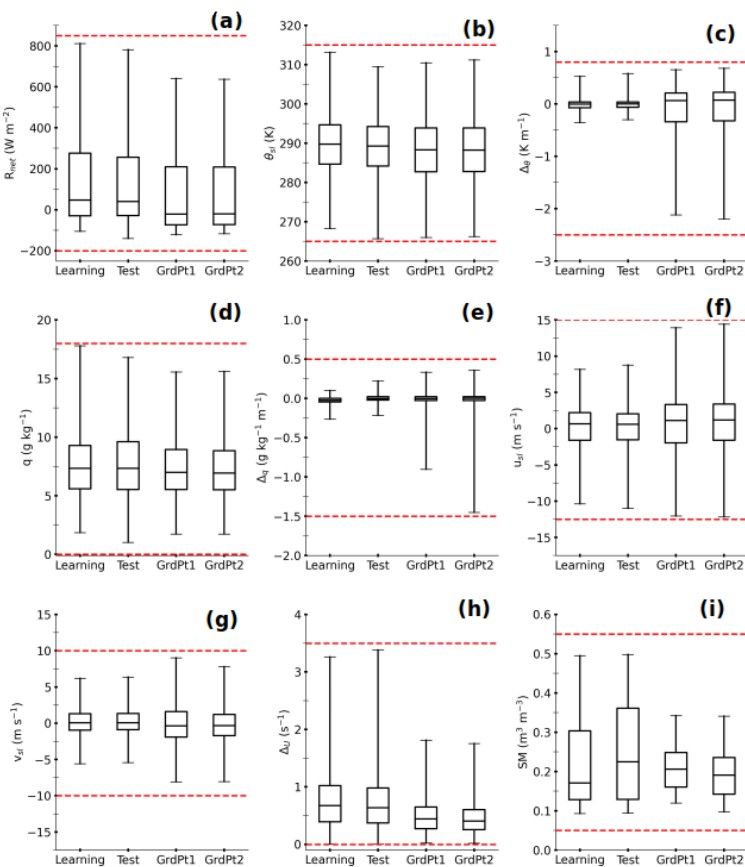

**Figure A1.** Box plots summarizing the interval ranges of the nine physical variables used as input to our MLP-based statistic model (Table 2). The corresponding data sets are indicated on the $x$-axis. The whiskers represent the minimum and maximum values of each data set. The horizontal dashed red lines indicate the extreme values used for scaling.







**Figure A2.** Composite monthly averages of observed sensible heat flux ($H$, a), latent heat flux ($LE$, b) and total turbulent heat flux ($H+LE$, c) for each year included in the observational data, calculated from the daily averages of half-hourly samples. The years of learning and test sets are in solid and dashed lines respectively. The thick lines correspond to the means on each subset and the error represents the 10 and $90^{th}$ percentiles.







**Figure A3.** Same as in Figure 10 but for GrdPt2.






**Figure A4.** Same as in Figure 11 but for GrdPt2.



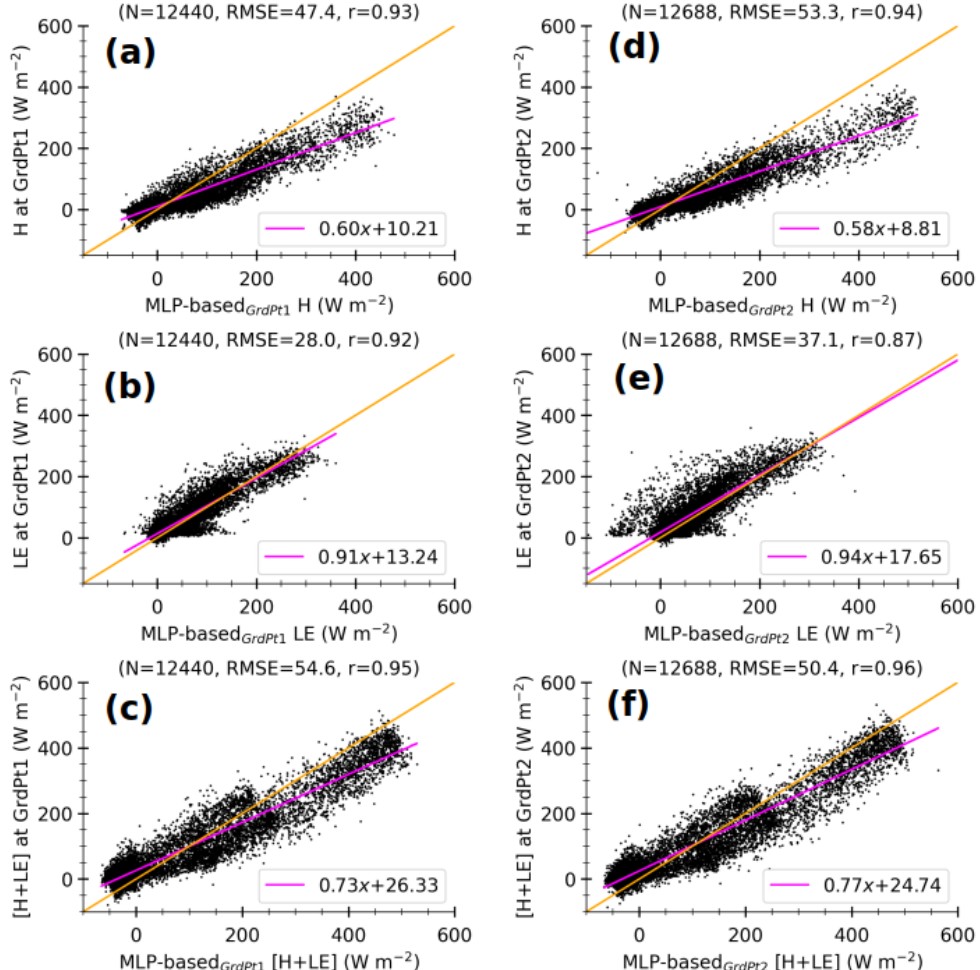

**Figure A5.** 3-hourly simulated sensible heat flux ($H$, a and d), latent heat flux ($LE$, b and e) and total turbulent heat flux ($H + LE$, c and f) at GrdPt1 (left panels) and GrdPt2 (right panels) against corresponding MLP-based estimates in simulated environment. All the selected timestamps from 01 January 2012 to 31 December 2016 are considered here. The values at the top of each panel correspond to the number of samples ($N$), Root-Mean-Square-Error ($RMSE$) and Pearson's correlation coefficient ($r$). The lines in magenta and orange represent the linear and ideal fits respectively.





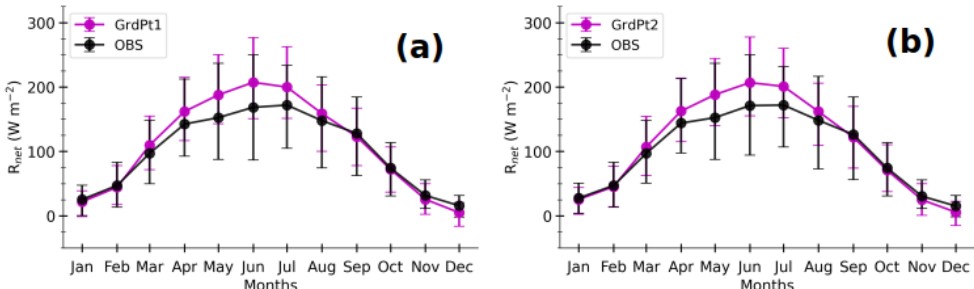

**Figure A6.** Composites monthly averages of simulated surface net radiative flux (magenta lines) at GrdPt1 (a) and GrdPt2 (b) and that of their respective observations at Météopole (black lines). They are computed as in Figure 12.

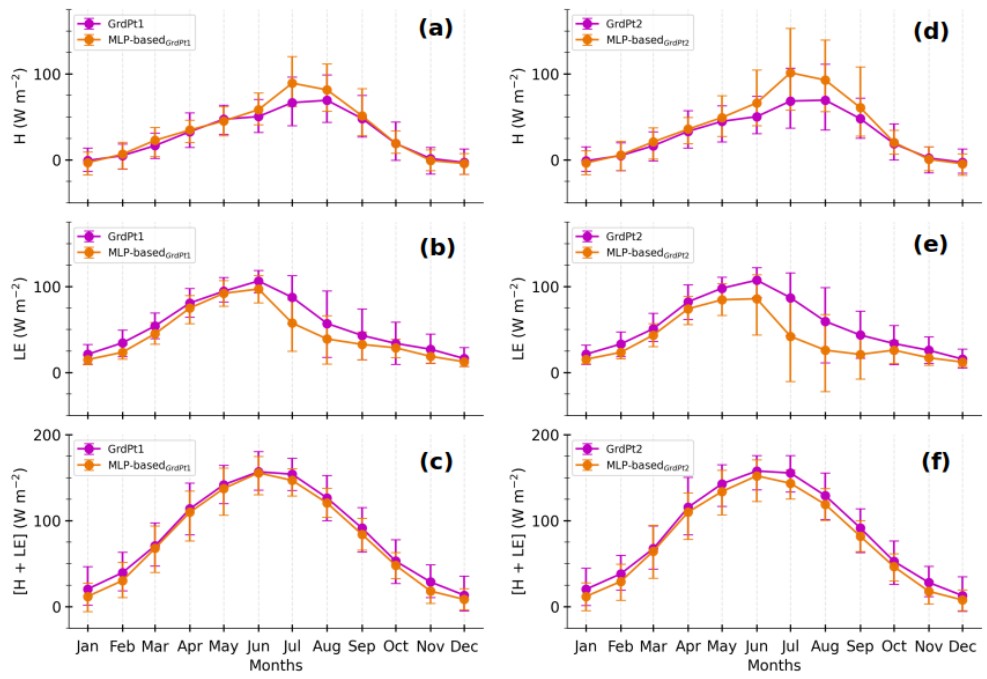

**Figure A7.** Composites monthly averages of simulated sensible heat flux ($H$, a and d), latent heat flux ($LE$, b and e) and total turbulent heat flux ($H + LE$, c and f) at GrdPt1 and GrdPt2 (magenta lines in left and right panels resp.) together with that of MLP-based estimates under simulated environment (orange lines resp.). The solid lines correspond to the means and the error bars represent the 10 and $90^{th}$ percentiles, calculated by gathering the daily averages of 3-hourly data. All the selected diurnal cycles from 01 January 2012 to 31 December 2016 are considered here.



*Author contributions.*    MZ, SB and MC elaborated the methodology with contributions from CM and LB. GC and SB provided the observational and climate simulation data respectively. MZ processed the data and prepared the paper with contributions from all the co-authors.

*Competing interests.*    The authors declare that they have no conflict of interest.

*Acknowledgements.*    The MOSAI (Model and Observation for Surface-Atmosphere Interactions) project was founded by the French Agence Nationale de la Recherche (ANR) under grant agreement no. 216875. The RegIPSL simulation was granted access to the HPC resources of
IDRIS under the allocation 2019-0227 attributed by GENCI (Grand Equipement National de Calcul Intensif). Thanks to ESPRI-IPSL and LATMOS IT department for access to the computational and storage infrastructures.



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
