# Peer review of "Using a data-driven statistical model to better evaluate surface turbulent heat fluxes in weather and climate numerical models: a demonstration study"

_EGUsphere, 2024_

## Author Comment (AC2)

General comment: Some of the grammar/language shall be revisited

We fully agree with this comment.

Specific comments:

Line 25: Mention also the first important source of bias.

The first source of bias in the numerical simulations is now mentioned. **P02, L25-26**: " *However, the representation of convection and surface processes are the two most important sources of systematic biases in numerical simulations (Zadra et al., 2018; Frassoni et al., 2023)."*

Line 74: In section 5

Correction done.

3.1:

Why not splitting the training datasets in each stability category (Stable, Neutral, Unstable) and different friction velocity ranges.

Splitting the training datasets based on surface stability and friction velocity is certainly a valid approach in some contexts. However, in our case, this suggests building separate statistical models for each category with not evenly distributing the data, with the risk of increasing the problem complexity (finding hidden layer topology) and imposing artificial discontinuities. Moreover, data-driven models trained on specific categories and ranges may struggle to generalize to unseen data, particularly when it falls near the boundaries of the defined categories. Therefore, we decide to build a single statistical model on the full range of conditions, using temperature and humidity vertical gradients ($\Delta_\theta$, $\Delta_q$) and vertical wind shear ($\Delta_U$) in the input parameters (Figure 5 and Table 2). In this way, the model learns to handle transitions between categories for better generalization in the fluxes' estimates.

Line 157: (ii) instead of (iii)

Correction done.

3.2:

Coarse time resolution (half-hour averages data may mask physical information for smaller time scales, at what time step the simulations were run?)

The simulation time step is of the order of a minute (90s), and output data have been archived at a coarser time resolution (3-hourly instantaneous or averages (depending on variables) and daily averages).

Lines 201-202:

The most conventional meteorological variables, such as T and RH at 2 m agl are also available. The lowest level as mentioned is within 20 m (pressure level akin to sigma coordinate which dynamically changes with time), so half eta-level is still higher than 2m. Any interpolation here?

Concerning numerical simulation, the atmospheric variables at conventional heights of observations (2 and 10 m agl) are diagnostic parameters. This means that they are output of the modeling system, calculated by interpolating between Earth's surface and lowest model level just above, using MOST stability functions. We avoid as much as possible the use of diagnostics that are based on a theory we want to assess.

Although, you mentioned this here Lines 303-305:

Moreover, the meteorological variables are directly taken at the first half-eta level (M=1, around 8 magl) instead of diagnostic variables as much as possible.

But 2m and 8m are quite far especially under stable conditions… Maybe, you need to comment on this..

The statement was modified in *P15, L307-309*:

"*Diagnostic variables derived from numerical simulation (T2m, RH2m) are susceptible to contain bias due to the interpolation technique (based on MOST) and the inconsistency of terrain elevation. To avoid these uncertainties, the meteorological variables are taken directly at the first half-eta level (M=1, around 8 m agl) as much as possible.*"

Line 202:

The data are stored at a temporal resolution of 3 hours …

How did you compare the instantaneous $3^{rd}$ hour snapshot with observations averaged in half-hour chunks?

Thank you for this comment which is very relevant because it points out a limitation of the model's evaluation (particularly for this method, but also in more classical approaches.) Indeed, simulated fluxes (turbulent and radiative) are not instantaneous but correspond to a time-centered mean over 3-hour, while temperature, humidity and wind data and vertical gradients are instantaneous. In the revised version, a section was added (Section 5.1) to clearly discuss the impact of the temporal resolution in both the comparison and the performance of the statistical model.

Line 204:

Meanwhile, the surface data, mostly provided by ORCHIDEE, consist of time-centred mean over a 3 hours window…

Specify what **surface data**…

The statement was completely revised (***P10, 205-212***).

Lines 209-213:

Could you show in Fig. 3 the grid-layout that shows the real observation geographic coordinate and the 2 nearest grid cells considered in the analysis.

Figure 3 now displays the grid mesh surrounding the geographical location of the Meteopole site (***P11***).

In this context, given the locality of space (dx=dy=20 Km) and the point observations collected from a tower at specific coordinates, have you done sensitivity analysis on the effect of spatial averaging?

We thank the reviewer for his comments. Our method seeks to better identify the land surface errors by performing the comparison within environment from the numerical model. The surface turbulent fluxes of the RegIPSL model are already derived from the weighted average of the parameters (roughness, albedo, etc.) over the different types of land cover present in grid cell in the same atmospheric forcing (***P10, 198-200***). Averaging the fluxes across grid cells would have been difficult to discuss, since environmental forcing and fluxes from different landscape composition are mixed. Hence, for sensitivity analyses, comparisons were initially made at the four grid cells closest to the observation site (see

figures below). Finally, the paper presents the results for the two closest grid cells, as the other two grid cells yield similar conclusions.

[Figure]

Composites monthly averages of simulated (lines in magenta) H, LE H+LE at the four nearest grid cells to the Météopole station, respectively, together with that of MLP-based estimates in simulated environment (lines in brown). The solid lines correspond to the means and the error bars represent the 10 and 90th percentiles, calculated by gathering the daily averages of 3-hourly data. All the selected diurnal cycles from 01 January 2012 to 31 December 2016 were considered here.

Also, any comments on the local dynamics forcings in the considered two grid cells compared to the observations? You considered the surface type aspect as a main criterion in selecting these 2 grid cells, but what about the topography effects and the local environmental dynamics (wind speed and directionality, temperature, …) when comparing outputs in these 2 grid cells to the observation's dataset at that specific grid location?

Our method is designed to evaluate the simulated fluxes within the simulated environment. The data-driven statistical model learns on measurements to provide potentially observed heat fluxes in the simulated environment. The benefit is that it is not necessary to have identical time series of environmental forcing between the simulation and observation. Aside from soil moisture (SM), the input variables at the two nearest grid cells exhibit a fairly similar range of variability (see Figure A1). The soil is slightly drier at the second grid cell (GrdPt2, Figure 3), likely because of different landscape composition, but the difference is weak compared to the variability obtained in the learning and test datasets.

3.3.1:

What's the criteria for selecting the training dataset?

The learning set was selected to adequately cover the inter-annual variability of turbulent fluxes with as many samples as possible.

Is it season independent?

Fortunately, learning data examples are well distributed across the four typical seasons (Figure 2b, *P09*).

Have you done any sensitivity analysis on choosing different dataset i.e., dataset convergence in terms of variability in the final weights and biases of MLP?

Yes, sensitivity analyses specifically focused on the choice of different datasets (including inputs of statistical model) and hyperparameters of MLP have been performed. The paper presents the experiment with the best results in terms of RMSE and correlation. The figures below show results of some sensitivity experiments performed for choosing input variables.

**➔ Tests on input variables**

[Figure]

**[12-8]**

$R_{net}, \theta_{sl}, \Delta_\theta, q_{sl}, \Delta_q, u_{sl}, v_{sl}, \Delta_U$ → H & LE

Correlation > 0.9

**These 8 input variables do not generalize correctly to unknown observed conditions.**

Learning = {2015, 2016, 2017, 2019, 2020}
Test = {2012, 2013, 2014, 2018, 2021}

**➔ Tests on input variables**

[Figure]

**[9-4]**

$R_{net}, \theta_{sl}, \Delta_\theta, q_{sl}, \Delta_q, u_{sl}, v_{sl}, \Delta_U, sm, d_{x,y}, h_{x,y}$ → H & LE

**Generalization is substantially enhanced with these 13 variables.**

Apprentissage = {2015, 2016, 2017, 2019, 2020}
Test = {2012, 2013, 2014, 2018, 2021}

3.3.2:

Lines 298-300: Define these variable acronyms especially:

Eventually, 4 trigonometric temporal coordinates are added for the description of seasonal (dx, dy) and diurnal (hx, hy) cycles.

All MLP inputs are defined in Table 2. The paragraph was modified, reference to Table is now at the beginning.

---

## Author Comment (AC3)

The results section could use some additional context and discussion, as well as revision for clarity. The connection between the background/motivation/introduction and the results gets lost at times. One small change that could assist this is a more specific naming of the cases. In section 5 in particular, I found myself frequently confusing the two MLP based fluxes and struggled at times to immediately understand the comparison being made. Perhaps assigning abbreviated case names (one for each of the four (or five if you count the different grid cells): estimated fluxes, observed fluxes, fluxes in the same environment, simulated fluxes) as well as text reminding what exactly they represent within the section could improve this clarity. The authors could use the naming already present in the figures in the text, for example, to have strong consistency. Language throughout sections 4 and 5 connecting back to the goals and motivation in the beginning of the paper would also help promote cohesion and make it easier to interpret.

Abbreviations are now used to clearly distinguish between observed fluxes (OBS), simulated fluxes (SIM), and MLP-based fluxes in the observed ($MLP_{OBS}$) and simulated environments ($MLP_{Grd}$). All these abbreviations were added in Figure 1. The comparison figures have been significantly updated to correspond with the schematic illustration in Figure 1. Accordingly, the discussion in section 4 and 5 have been revised for clarity.

Finally, there are a few differences between the simulations and the tower observations (and the MLPs based on them) that hinder comparison. While the authors do not necessarily avoid talking about them, the discussion on them is scattered throughout and could be enhanced with a more detailed and focused discussion. In particular answering:

- How does the mismatch of temporal resolution (30 min vs 3 hour) affect the results, particularly since we would not expect fluxes to be stationary over 3 hours (especially during the mornings/evenings)?

The mismatch in temporal resolution between observational and model data (30 minutes vs. 3 hours) can obviously introduce some challenges, especially since fluxes are not typically stationary over a 3-hour period, particularly during transition between stable and unstable regimes. In the revised version, a section was added (Section 5.1) to clearly discuss the impact of the temporal resolution in both the comparison and the performance of the

statistical model. We initially applied this approach to an already existing climate simulation, and in future steps, it will be used for simulations performed in the framework of the MOSAI project.

- How effectively can we compare between 20 km grid cells with different (and heterogeneous) land cover and a tower with ~4m agl flux readings which is likely only reading a small area from a grassland?

This study aimed to develop an evaluation method that firstly addresses the disparity in environmental conditions between simulation and observation. Future work in MOSAI project will address other challenges that hinder reliable evaluation of land surface scheme. These challenges include, non-closure of SEB in observed fluxes and the representativeness of in-situ measurement at the coarser horizontal resolution of numerical model. An enhanced observation period of one-year has been conducted for this purpose, by adding additional flux measurements over different types of land use around the main site. A general paper presenting the MOSAI project will be submitted soon. The conclusion has been modified for clarity (***P33, L613-620***).

- What is lost by neglecting the soil/surface temperature? Those should have a strong correlation with the sensible heat flux in particular.

Surface temperature and sensible heat flux are indeed closely related. However, due to a significant amount of missing data (~40%), we chose not to include surface temperature when deriving the input variables for the statistical model.

Table below shows matrix of correlation coefficient between sensible heat flux (H), net radiative flux (Rnet) and potential temperature $\theta_{surf}$ (calculated with surface pressure and surface brightness temperature) for all available half-hourly observational data (between July 2015 to December 2022) .

| | H | $R_{net}$ | $\theta_{surf}$ |
|---|---|---|---|
| H | - | 0.912880 | 0.721587 |
| $R_{net}$ | 0.912880 | - | 0.721798 |
| $\theta_{surf}$ | 0.721587 | 0.721798 | - |

It can be observed that H is more strongly correlated with $R_{net}$, while both $R_{net}$ and H show similar correlations with $\theta_{surf}$. As $R_{net}$ is one of the input variables, there is no significant loss of critical information regarding the variability in H and even LE. A new sentence was added in the revised version (***P06, L133***) :

*Sensitivity analysis indicates no significant loss of key information concerning the variability in H and LE.*

**Technical Corrections have been done**

---

## Author Response (AR2)

**Interactive comment on "Using a data-driven statistical model to better evaluate surface turbulent heat fluxes in weather and climate numerical models: a demonstration study (https://doi.org/10.5194/egusphere-2024-568)"**

Dear reviewers,

Thank for your helpful suggestions, which led to significant improvements of this paper.

**The syntax error in Figure 1 has been corrected, and a more detailed caption has been added (Page 5).**

➔ **Tests on input variables**

[Figure]

R$_{net}$, θ$_{sl}$, Δ$_θ$, q$_{sl}$,
Δ$_q$, u$_{sl}$, v$_{sl}$, Δ$_U$,
sm, d$_{x,y}$, h$_{x,y}$

**[9-4]**

H & LE

Apprentissage = {2015, 2016, 2017, 2019, 2020}
Test = {2012, 2013, 2014, 2018, 2021}

Generalization is substantially enhanced with these 13 variables.